# Key Strategies to Optimize Outcomes in Mild-to-Moderate Ulcerative Colitis

**DOI:** 10.3390/jcm9092905

**Published:** 2020-09-08

**Authors:** Virginia Solitano, Ferdinando D’Amico, Gionata Fiorino, Kristine Paridaens, Laurent Peyrin-Biroulet, Silvio Danese

**Affiliations:** 1Department of Biomedical Sciences, Humanitas University, Pieve Emanuele, 20090 Milan, Italy; virginia.solitano@humanitas.it (V.S.); ferdinando.damico@humanitas.it (F.D.); gionataf@gmail.com (G.F.); 2Department of Gastroenterology and Inserm NGERE U1256, University Hospital of Nancy, University of Lorraine, 54500 Vandoeuvre-lès-Nancy, France; peyrinbiroulet@gmail.com; 3Department of Gastroenterology, IBD Center, Humanitas Clinical and Research Center, IRCCS, Rozzano, 20089 Milan, Italy; 4Ferring International Center, 1162 St-Prex, Switzerland; Kristine.Paridaens@ferring.com

**Keywords:** ulcerative colitis, mild to moderate, 5-aminosalicylate, budesonide multimatrix system, adherence, personalized medicine, treat to target

## Abstract

Mesalamine (5-ASA) is the mainstay therapy in patients with mild-to-moderate active ulcerative colitis (UC). However, non-adherence to therapy and practice variability among gastroenterologists represent long-standing barriers, leading to poor outcomes. Additionally, targets to treat in UC are increasingly evolving from focusing on clinical remission to achieving endoscopic and histological healing. To date, systemic steroids are still recommended in non-responders to 5-ASA, despite their well-known side effects. Importantly, with the advent of new therapeutic options such as oral corticosteroids with topical activity (e.g., budesonide multimatrix system (MMX)), biologics, and small molecules, some issues need to be addressed for the optimal management of these patients in daily clinical practice. The specific positioning of these drugs in patients with mild-to-moderate disease remains unclear. This review aims to identify current challenges in clinical practice and to provide physicians with key strategies to optimize treatment of patients with mild-to-moderate UC, and ultimately achieve more ambitious therapeutic goals.

## 1. Introduction

Ulcerative colitis (UC) is a chronic, immune-mediated inflammatory bowel disease (IBD), characterized by recurrent flares and periods of remission [1]. It generally affects the rectum and extends proximally to other colonic segments, causing abdominal pain and bloody diarrhoea, thereby leading to impairments in quality of life and work productivity combined with an increased risk of colorectal cancer [2,3,4,5]. Medical treatment is the mainstay of UC management and is generally contingent on colonic involvement and disease severity (mild, moderate, or severe) [6]. Colonic involvement is classified as proctitis (E1), left-sided colitis (E2), or extensive colitis (E3) according to the Montreal Classification [7,8]. On the other hand, the Truelove–Witts criteria categorize UC severity as mild, moderate, or severe taking into account the number of bowel movements per day, temperature, heart rate, haemoglobin (Hb) level, and erythrocyte sedimentation rate [9]. Left-side localization with mild-to-moderate activity (less than 6 bowel movements per day, without constitutional symptoms and features of high inflammatory activity) is the most common disease presentation at diagnosis, accounting for 40% of cases [10]. Current guidelines recommend oral and/or topical 5-aminosalicylic acid (5-ASA; mesalamine) as first-line medication for induction and maintenance therapy in mild-to-moderate UC, reserving oral systemic steroids for patients who are either intolerant or not adequately controlled with 5-ASA [11]. However, despite their effectiveness in inducing remission, the use of oral corticosteroids is limited by their well-known adverse effects (AEs) [12]. As therapeutic goals are becoming more ambitious, including not only symptomatic but also endoscopic and histological remission [13], it is vital for clinicians to optimize the application of currently available treatments and to promptly identify patients who might benefit from escalating to a more intensive therapy [14]. The purpose of this review is to identify drawbacks and challenges in the management of mild-to-moderate UC, with the aim of providing key effective strategies to support physicians in daily practice in the 2020 clinical scenario.

## 2. Current Recommendations in Mild-to-Moderate UC

The 2017 European Crohn’s and Colitis (ECCO) guidelines recommended a mesalamine 1-g suppository once daily as first-line therapy in proctitis [11]. An aminosalicylate enema ≥1 g/day combined with oral mesalamine ≥2.4 g/day should be administered in patients with left-sided and extensive UC [11]. In those who did not respond to mesalamine, systemic corticosteroids were considered an appropriate option despite the known AEs associated with steroid therapy (e.g., mood and sleep disorders, infections, diabetes, hypertension, and bone disease) [11,15,16]. To overcome this limitation, the use of budesonide multimatrix system (MMX) was proposed. MMX technology is a drug delivery system that provides a prolonged and consistent release of the active substance throughout the colon [17]. Budesonide MMX was found to be effective and safe in patients with mild-to-moderate UC and is currently recommended in patients with mild-to-moderate left-sided colitis who are intolerant of or refractory to 5-ASA [11]. With regard to maintenance therapy, for which the goal is to ensure prolonged clinical and endoscopic steroid-free remission [11], mesalamine is the main drug for this indication, including a dosage of 2 g/day for oral and 3 g/week for rectal formulations [11]. In patients with early or frequent relapses during 5-ASA therapy, or in case of 5-ASA intolerance or steroid dependence, thiopurine monotherapy (azathioprine or 6-mercaptopurine) are recommended for maintenance therapy in patients with mild-to-moderate UC [11].

## 3. Drawbacks and Challenges in Mild-to-Moderate UC Management

It should be emphasized that therapeutic targets in UC are changing from mucosal healing to histological healing [18]. This target should not only be realized in patients with moderate-to-severe disease but should also be applied to patients with mild-to-moderate UC [18]. Thiopurines have a slow mechanism of action requiring several weeks to reach their therapeutic effect, have a limited impact in the induction of histological remission in UC patients, and are associated with a high number of serious AEs (e.g., pancreatitis, lymphoma, and non-melanoma skin cancer) [19,20,21]. This risk/benefit ratio is not acceptable in subjects with mild-to-moderate disease and raises several doubts about the thiopurines’ use in this setting. However, about half of the patients do not achieve sustained remission after optimization with 5-ASA and budesonide [22]. Thus, immunomodulators are still necessary to avoid the excessive use of systemic steroids and to maintain corticosteroid-free remission [10]. Another drawback is related to patients’ follow-up. In fact, many patients with mild-to-moderate UC are monitored with proctosigmoidoscopy rather than with pancolonoscopy. A study by Colombel and colleagues investigated the correlation between proctosigmoidoscopy and pancolonoscopy in assessing endoscopic disease activity [23]. A high degree of correlation was found between the two procedures when mucosal healing was defined as Mayo = 0 (κ = 0.95; r = 0.95). On the other hand, when a higher threshold was used (Mayo ≤ 1), almost twice the percentage of endoscopic remission was found with proctosigmoidoscopy compared to that with pancolonoscopy (14% vs. 8%), suggesting a diagnostic overestimation [23]. This finding is very important since an inaccurate assessment could lead to disease undertreatment and inferior outcomes. For this reason, it has been hypothesized that non-invasive markers such as faecal calprotectin (FC) could be useful for the follow-up of UC patients, as FC is closely associated with the endoscopic and histological activity of disease [24]. A randomized study by Lasson et al. compared the outcomes of UC patients who underwent 5-ASA optimization, based on FC values, with standard care [25]. The number of relapses during an 18-month follow-up was significantly higher in patients with FC > 300 µg/g who remained on standard care compared to that in those undergoing dose optimization in the experimental group (57.1% vs. 28.6%, *p* < 0.05) [25]. Similarly, a randomized study by Osterman et al. showed that patients treated with high-dose 5-ASA (4.8 g/day) achieved a higher rate of remission with FC < 50 µg/g compared to patients treated with low-dose 5-ASA (2.4 g/day) (26.9% vs. 3.8%, *p* = 0.0496), suggesting that the measurement of FC could be useful for improving disease assessment and, thereby, to avoid repeated endoscopic tests that are poorly tolerated by patients [26]. However, FC measurement has some limitations as there are no standardized procedures related to its performance and a commonly accepted cutoff is not available [27]. In this context, bowel ultrasound could be an adjunct to FC as it is a non-invasive, inexpensive, and well-tolerated method with a high accuracy for the detection of endoscopic disease activity (sensitivity 0.71, specificity 1.00) [28].

Non-adherence to therapy is another well-known barrier when dealing with UC patients since it leads to negative long-term outcomes and increased costs [29,30]. Several studies reported an increased risk of clinical relapse, colorectal cancer, and healthcare costs due to non-adherence [31,32,33]. Compliance is an issue with all IBD medications [34,35]. Shale et al. identified three-times per day dosing and full-time work as independent risk factors for partial compliance, drawing attention to the need for optimized maintenance-drug regimens [36]. Besides complicated dosing regimens and an onerous pill burden, a systematic review conducted by Kane et al. suggested that the fear of side effects was among the main reasons for non-adherence [37]. Moreover, age <40 years also contributed to non-adherence [36,38] and topical therapy with suppositories or enemas was more likely to be associated with non-adherence than oral therapy (68% vs. 40%, *p* = 0.001, odds ratio (OR) 0.25, confidence interval (CI) = 0.11–0.60) [38]. Finally, variability in medical practice among general gastroenterologists and IBD experts constitutes an obstacle for optimal management of mild-to-moderate UC [39]. A large survey of 700 gastroenterologists highlighted some discrepancies between European recommendations and clinical practice [40]. For instance, less than 50% of gastroenterologists prescribed a combination of oral and topical 5-ASA for distal colitis, and once-daily dosing of mesalamine was recommended by about half of the respondents [40], which indicates that at least half of the clinicians were not adhering to the guidelines in each context.

## 4. Key Strategies to Improve the Management of Mild-to-Moderate UC

### 4.1. 5-ASA Optimization

#### 4.1.1. Once-Daily (OD) Dosing

Efficacy data on different doses and formulations of mesalamine are summarized in Table 1. Simplifying the intervals and dosage of a drug regimen proved to be an efficacious strategy [41,42,43,44]. After randomizing UC patients to prolonged-release oral mesalamine 2 g OD or 1 g twice daily, Dignass et al. reported better remission rates (70.9% vs. 58.9%, *p* = 0.02) and self-reported adherence (measured through a 100 mm visual analogue scale (VAS)) in the former group (95.6 vs. 93.8 mm, *p* = 0.033) [41]. Conversely, in a meta-analysis of randomized controlled trials including 738 patients, no significant differences were observed in terms of remission rates between OD and conventional 5-ASA dosing regimens (relative risk (RR) = 0.95, 95% CI = 0.82–1.10) [42]. These data were confirmed by the MOTUS trial, where OD prolonged-release mesalamine (4 g) was comparable with 2 g twice-daily dosing for induction of clinical and endoscopic remission, defined as UC disease activity index (UCDAI) ≤1 (52.1% vs. 41.8%, *p* = 0.14) [43]. Moreover, although compliance was not superior in the OD arm, 5-ASA 4 g in a single dose was well tolerated compared with multi-dosing regimens [43]. In line with these findings, D’Haens et al. showed that 3.2 g mesalamine OD was comparable to a twice-daily regimen in inducing clinical and endoscopic remission at week 8 (22.4% vs. 24.6%, respectively, non-inferiority *p* = 0.005) [44]. Not surprisingly, the single-dose schedule was also associated with higher patient satisfaction, when compared to multiple daily dosing [44]. A multicentre, randomized, active-control trial enrolled 1023 UC patients to determine the non-inferiority of 5-ASA OD compared to a standard regimen. Interestingly, the experimental dosage was reported to be as effective as the control group in maintaining clinical remission (85.4% vs. 85.4%, 95% CI = −4.6 to 4.7), but more satisfied subjects were identified in the OD arm than in the twice-daily dosing group after 12 months of therapy, based on the Patient Satisfaction and Preference Questionnaire (58.3% vs. 45.4%, *p* = 0.117) [45].

MMX mesalamine is an OD formulation that delays and extends the release of active drug throughout the colon, potentially leading to improved patient compliance [55]. The efficacy of MMX 5-ASA in patients with mild-to-moderate UC was assessed in two randomized phase III studies [46,47]. Kamm et al. reported significantly higher clinical and endoscopic remission rates at week 8 in MMX 5-ASA 2.4 and 4.8 g/day compared to placebo (40.5% and 41.2% vs. 22.1%, with *p* = 0.01 and *p* = 0.007, respectively) [46]. Likewise, greater clinical and endoscopic remission rates at week 8 were found by Lichtenstein et al. in patients receiving MMX 5-ASA 2.4 g twice daily or MMX 5-ASA 4.8 g/day compared to placebo (34.1% and 29.2% vs. 12.9% placebo, *p* < 0.01) [47]. Focusing on adherence, in a phase IV open-label study, called SIMPLE (Strategies in maintenance for patients receiving long-term therapy), almost 80% of patients were ≥ 80% adherent to MMX mesalamine after 12 months of follow-up [48]. Of note, patients who were ≥ 80% adherent had a lower rate of disease relapse at 6 and 12 months compared to patients with < 80% adherence (20.6% and 31.2% vs. 36.1% and 52.5%, with *p* = 0.05 and *p* = 0.01, respectively) [48]. In addition, an open-label prospective trial including 717 patients with active mild-to-moderate UC showed that MMX 5-ASA was associated with significant improvements in patient-reported outcomes (PROs) measuring health-related quality of life (SF-12v2^®^ Health Survey and Short Inflammatory Bowel Disease Questionnaire) and work-related outcomes (Work Productivity and Activity Impairment questionnaire) (*p* < 0.001 for all comparisons) [56].

#### 4.1.2. Combination of Oral and Rectal 5-ASA

The role of oral and topical combination 5-ASA therapy needs to be clarified [57]. According to the most recent European guidelines, topical 5-ASA monotherapy is recommended in mild-to-moderate proctitis, whereas combination is the first-line therapy for inducing remission in left-sided and extensive colitis [11]. Importantly, two randomized controlled trials revealed that combined oral and rectal 5-ASA was an effective strategy in patients at high risk of relapse during maintenance treatment [49,50]. An Italian study recruited patients with a history of two or more relapses in the previous year reporting a relapse rate of 39% in the combined treatment group vs. 69% in those receiving oral 5-ASA alone (*p* = 0.036) [49]. In this study, oral 5-ASA was supplemented with topical administration on Saturday and Sunday [49]. Similarly, a study conducted by Yokoyama et al. found lower relapse rates in the arm treated with both 5-ASA formulations than in the oral 5-ASA group (18.2% vs. 76.9%, respectively; multivariate hazard ratio (HR) 0.19, 95% CI, 0.04–0.94) [50]. Based on these two studies, a systematic review recently confirmed that combination therapy with oral and topical 5-ASA was superior to oral 5-ASA alone for maintenance of remission (RR, 0.45 (0.20–0.97)) [39]. Thus, since adherence to rectal treatment is often suboptimal [38], intermittent administration of topical therapy constitutes a valid option, particularly in patients at high risk of relapse [48].

#### 4.1.3. Increasing Dose of 5-ASA

High-dose mesalamine (4.8 g/day) should be considered in patients with moderate UC, as demonstrated in the ASCEND I and II trials [51,52]. Hanauer et al. compared the efficacy and safety of mesalamine 4.8 g/day with that of mesalamine 2.4 g/day. In ASCEND I, a higher overall improvement at week 6 was detected in patients with moderate UC treated with mesalamine 4.8 g/day compared with that for those receiving mesalamine 2.4 g/day (72% vs. 57%, *p* = 0.0384) [51]. In ASCEND II, treatment success was achieved in more patients treated with high dosage than in those treated with standard dosage (72% vs. 59%, respectively, *p* = 0.036), and the rate of serious AEs was comparable between the two groups (1.4% vs. 0.8%), confirming that doubling the dose was an effective and safe strategy [52]. In line with these findings, a more recent trial evaluating patients with moderately active UC receiving 4 versus 2.25 g/day for 8 weeks reported a greater improvement in UCDAI score in the former group (UCDAI score change −3.0 (95% CI: −3.8 to −2.3) vs. −0.8 (95% CI: −1.8 to 0.1)), suggesting that high-dose mesalamine (≥4g/day) should be considered at the outset in this subgroup of patients [53]. An economic study further supported the use of a double dose of 5-ASA in patients with moderate UC due to its overall cost-effectiveness [58]. Using an analytical model, Buckland et al. argued that the increased drug expenses associated with 12-week high-dose mesalamine (£2474 vs. £2382 per patient) were outweighed by cost-savings in other areas, such as the potential reduction in hospitalization rates (20% vs. 22%) [58]. As regards maintenance therapy, a Cochrane meta-analysis revealed that a high dose of mesalamine maintenance therapy (≥2 g/day) could decrease the number of disease relapses in high-risk UC patients (relative risk (RR) 0.73, 95% CI 0.60–0.89) [59]. An Italian trial compared two dosage regimens of oral mesalamine (4.8 and 2.4 g/day) in UC patients who had experienced at least 3 relapses within one year [54]. Interestingly, significant differences in terms of maintenance remission rates were observed between the two arms, especially in patients younger than 40 years (90.5% vs. 50%, *p* = 0.0095) and/or with extensive disease (90.9% vs. 46.7%, *p* = 0.0064) [54]. Although few data are available, it seems likely that patients who required higher doses of oral 5-ASA for the induction of remission might also benefit from this dosage during the maintenance phase [11,60].

### 4.2. Budesonide MMX Integration in the Therapeutic Armamentarium

New steroid formulations, such as budesonide MMX, have been developed to address the problem of toxicity linked to systemic corticosteroids [52]. MMX technology allows the active substance to be released uniformly along the colon [61], thereby minimizing its absorption into the systemic circulation [62]. The efficacy and safety of budesonide MMX in mild-to-moderate UC were first assessed by the CORE randomized controlled trials [63,64,65]. Budesonide MMX 9 mg was significantly more efficacious than placebo in inducing clinical and endoscopic remission (17.9% vs. 7.4%, *p* = 0.0143 in CORE I and 17.4% vs. 4.5% in CORE II) [63,64]. As regards safety and tolerability, a pooled analysis of 5 clinical studies reported similar rates of AEs across three treatment groups (27.1%, 24.8%, and 23.9% in patients receiving budesonide MMX 9 mg, 6 mg, or placebo, respectively) [66]. Moreover, this second-generation steroid was not associated with an increased rate of glucocorticoid-related side effects compared to placebo (9.6% vs. 9.8%) [66]. Another trial investigated the role of budesonide MMX as add-on therapy in patients inadequately controlled by oral mesalamine therapy (≥2.4 g/day for at least 6 weeks) [67]. Patients were randomly assigned to receive OD budesonide MMX or placebo, and significant differences were reported between treatment arms in terms of clinical and endoscopic remission (13% vs. 7.5%, *p* = 0.049), endoscopic remission alone (20% vs. 12.3%, *p* = 0.025), and histological remission (27% vs. 17.5%, *p* = 0.016) after 8 weeks [67]. A network meta-analysis of 15 randomized controlled trials corroborated these data showing that budesonide MMX 9 mg and mesalamine >2.4 g/day were equivalent for inducing clinical and endoscopic remission (OR = 0.97, 95% CI: 0.59–1.60) [68]. Additionally, two observational cohort studies provided evidence in a real-world setting [69,70]. In a retrospective multicentre study including 82 patients with mild-to-moderate UC, 50% of patients treated with budesonide MMX achieved clinical remission and a further 9.8% had a clinical improvement [69]. Interestingly, most patients (54/82 = 66%) received budesonide MMX as an add-on drug suggesting that combination therapy might be a valid therapeutic approach [69]. In a multicentre prospective study by Danese et al., clinical outcomes were improved among patients receiving combination therapy (cohort 1 and 2) compared to that among those receiving budesonide MMX monotherapy (cohort 3) [70]. Clinical benefit, defined as a reduction of ≥3 points in UCDAI score at the end of treatment, was achieved in 64.3%, 62.1%, and 33.3% in cohorts 1, 2, and 3, respectively [70]. Notably, clinical benefit was obtained regardless of the time of budesonide MMX addition after attempting 5-ASA dose optimization [70]. Moreover, the safety data were reassuring, as budesonide MMX was well tolerated and less than 25% of patients reported AEs [70]. Despite these promising data (Table 2), the position of budesonide MMX among induction agents remains controversial [71]. According to ECCO guidelines, budesonide MMX should be used in left-side colitis after 5-ASA failure, whereas the American Gastroenterological Association (AGA) suggests adding budesonide MMX, regardless of disease extent [60]. Based on a systematic review and network meta-analysis of randomized controlled trials supporting the efficacy of budesonide MMX in patients with moderate UC, it can legitimately be considered as a first-line therapy or after 5-ASA optimization failure.

### 4.3. Patient Stratification for Earlier Intervention

Therapeutic targets in IBD have shifted from controlling symptoms to the adoption of a treat-to-target strategy [72], where not only clinical remission but also endoscopic and histological healing are considered desirable objectives to achieve [73]. Reducing inflammatory burden at an early stage constitutes an integral part of the treat-to-target approach, which involves choosing initial therapy according to the risk of disease progression, assessing baseline features of disease, monitoring the progress, and optimizing treatments in order to reach the agreed goals (Figure 1) [72]. In this new clinical scenario, a prompt individualized treatment during early disease stages is required to prevent irreversible bowel damage and to improve long-term outcomes [74]. 

Of note, UC is considered by most physicians as a less progressive disease compared to Crohn’s Disease, explaining the reluctant attitude to introduce more effective treatments earlier in the course of the disease and the risk of undertreatment [75,76]. Subsequently, selecting those patients who are at risk of disease progression and treating them accordingly is becoming of crucial importance [77]. Particularly among patients with moderate UC, attention should be paid to those with negative prognostic factors and those who might benefit from a more intensive initial therapy [78]. Predictors of a more aggressive disease course are young age at diagnosis, extensive disease, extra-intestinal manifestations, presence of endoscopic signs of disease activity (e.g., ulcers), and elevated inflammatory biomarkers (FC and C-reactive protein) [60,79]. In patients with several “red flags”, the early use of biologics could be indicated to prevent occurrence of negative outcomes and disease progression. Interestingly, the UC Clinical Care Pathway suggested the use of biological agents as first-line therapy not only in patients with severe disease activity but also in those with moderate activity who were at high risk of colectomy, encouraging a flexible and personalized approach to making therapeutic decisions [80]. Accordingly, we propose the consideration of earlier exposure to biologics or small molecules on a case-by-case basis, relying on a careful evaluation of both negative prognostic factors at diagnosis and the potential risks of overtreatment and AEs. Although decisions should be based on a personalized approach and more sophisticated clinical decision support tools are needed, modifying the natural history of the disease and preserving intestinal functionality should be the guide in any therapeutic choice. 

### 4.4. Shared Decision-Making and Patient Involvement

Shared decision-making between clinicians and patients has been proposed as an effective strategy to increase patients’ adherence, satisfaction, and quality of life [81]. Shared decision-making incorporating the patient’s health beliefs and concerns is highly recommended as an integral part of a personalized therapeutic approach [82]. A prospective study by Pedersen et al. enrolled 95 UC patients who had been non-adherents or non-responders to previous rectal and/or oral 5-ASA therapy in order to evaluate a self-managed, individualized 5-ASA treatment approach [83]. By employing a web-based approach, which allowed for subjective and objective monitoring, treatment dosage and duration were individualized [83]. Interestingly, treatment adherence was high (baseline VAS median of 80 vs. 100 mm at the end of the study, *p* = 0.001) and disease activity significantly improved in the short term (total inflammatory burden score (TIBS) 6.7 vs. 2.4, *p* = 0.001) [83]. Another randomized controlled trial compared standard care with an adherence-enhancing strategy, which involved education and motivation of UC patients together with practical reminders and options for simplified dosing regimens [84]. In the tailored-treatment arm, adherence was significantly higher than that in the control group (76% vs. 32%, *p* < 0.001) [84]. This trial reinforced the hypothesis that empowering patients, through education and enhanced comprehension of the instructions for the appropriate use of medication, together with an increased awareness of the likely consequences of non-adherence, significantly improved patient treatment adherence [85]. Along these lines, in 2018 a consensus of an international panel of experts placed a high value on patients’ active involvement in their disease management [86], and AGA recommendations included the possibility for patients with mild-to-moderate UC to choose oral 5-ASA over rectal therapy [60].

## 5. A Proposed Algorithm for Practical Guidance 

Based on the body of evidence, we believe that it is time to revise current treatment algorithms towards a tailored management of mild-to-moderate UC (Figure 2). In patients with mild UC, 5-ASA should be the recommended first-line therapy, and the regimens should depend on the extent of disease proliferation according to international guidelines [11,60]. To improve adherence rates, treatment decisions should be driven by patient’s preferences, for example, by opting for OD dosing or prescribing oral therapy if suppositories and enemas are not well accepted [37]. When dealing with patients with moderate disease at diagnosis, a prompt evaluation of prognostic factors should be undertaken [78]. In those at low risk of colectomy, high-dose mesalamine (≥4 g/day) or budesonide MMX with 5-ASA could be a rational strategy regardless of the extent of disease. The choice between these two strategies could be based on their tolerability for patients since a recent network meta-analysis of induction therapy trials found that high-dose mesalamine was better tolerated than budesonide MMX [14]. Although future research on the juxtaposition of budesonide MMX among induction therapies is needed, this second-generation corticosteroid could be a valid treatment not only as an add-on therapy but also in first-line coadministration with 5-ASA [14]. In patients non-responsive to budesonide MMX or those with moderate UC and negative prognostic factors, systemic corticosteroids could be appropriate, despite their poor safety profile [87]. Attention should be paid to the deleterious and disastrous consequences of a wrong use of systemic steroids. In order to prevent the vicious circle of on-and-off steroids, which is not the appropriate way to manage the inflammatory burden of disease, the recommended regimen of systemic corticosteroids should entail prednisolone 40 mg/day (or corticosteroid dose equivalents) for 1 week, lowering the daily dose by 5 mg each week, without exceeding the duration of an 8-week course. At the same time, shorter courses (<3 weeks) and ineffective doses of prednisolone (≤15 mg/day) should be avoided [11]. Identification of the subset of patients with moderate UC who could benefit from early introduction of biological therapy could be warranted, and evidence regarding early intervention with biological agents and/or small molecules in UC is awaited [18]. For the maintenance of mild UC, standard therapy with 3 g/week topical 5-ASA and 2 g/day oral 5-ASA represents an effective strategy [11]. High-dose mesalamine (>2 g/day) with or without rectal 5-ASA should be initially considered in patients with mild-to-moderate disease who are at high risk of disease progression or recurrence, while biological drugs or small molecules could be a valid second-line option in case of 5-ASA optimization failure. This hypothesis is in line with the well-established top–down strategy and could lead to the achievement of more ambitious goals for disease control already seen in patients with moderate-severe disease [88]. 

Furthermore, in patients who have experienced relapse while taking 5-ASA compounds and in those who are steroid dependent, thiopurines are still recommended during the maintenance phase, due to their steroid-sparing effect [11]. They could also be used as a component of combination therapy with anti-tumor necrosis factor (TNF) to reduce their immunogenicity and to increase the probability of response in the short term [89]. However, as mentioned above, the use of thiopurines is affected by its limited therapeutic effect and its well-known toxicity, so that with the advent of new effective biological agents and small molecules, the positioning of thiopurines in the therapeutic algorithm might change [90].

## 6. Conclusions

Non-adherence to medications, side effects of systemic steroids, and practice variability among physicians represent barriers constraining the best therapeutic management of mild-to-moderate UC patients. Greater attention should be paid to the follow-up of these patients including not only the use of pancolonoscopy but also non-invasive tests such as FC and bowel ultrasound. With regard to treatment, it is clear that case-by-case decisions, taking into account patients’ prognostic factors and individual needs, are warranted. Furthermore, 5-ASA optimization should be routinely employed, while an OD regimen could improve patient compliance. Moreover, combining the oral and topical route or doubling the dose of mesalamine should be considered in patients at high risk of relapse during maintenance, whereas integrating budesonide MMX into the therapeutic armamentarium constitutes a valid option for the induction of clinical remission. In conclusion, studies evaluating the efficacy and safety of biologics and small molecules in patients with moderate UC are needed to determine whether these drugs should be incorporated into the therapeutic regimen for the realization of evolving therapeutic goals.

## Figures and Tables

**Figure 1 jcm-09-02905-f001:**
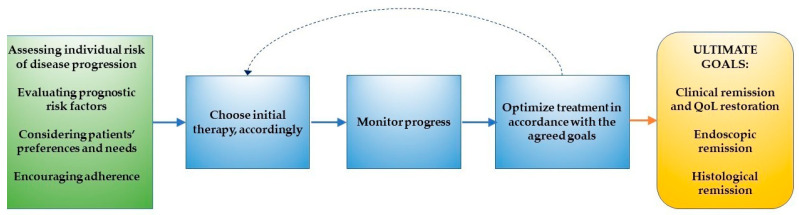
Steps to be taken in order to achieve evolving therapeutic goals according to a treat-to-target approach. QoL: quality of life.

**Figure 2 jcm-09-02905-f002:**
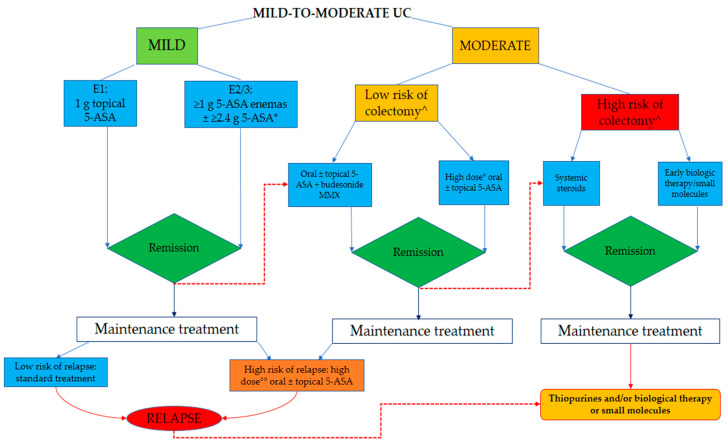
Proposed algorithm for the management of mild-to-moderate UC. *: OD dosing and MMX mesalamine are valuable options. ^: according to prognostic factors: young age at diagnosis, extensive disease, extra-intestinal manifestations, presence of endoscopic signs of disease activity (e.g., ulcers), and elevated inflammatory biomarkers. °: ≥4 g/day; °°: >2 g/day.

**Table 1 jcm-09-02905-t001:** Studies reporting the efficacy of different doses and formulations of mesalamine in patients with mild-to-moderate ulcerative colitis.

Author (Year)	Study Design	Number of Patients	Study Arms	Primary Outcome	Results	Conclusions
Dignass et al. (2009) [41]	Randomized non-inferiority trial	362	5-ASA (2 g) OD 5-ASA (1 g) BD	1-yr. remission rates (UCDAI score <2)	70.9% 58.9% (*p* = 0.024)	Prolonged-release oral 5-ASA 2 g once daily is associated with better remission rates
Flourie et al. (2013) [43]	Randomized non-inferiority trial	206	5-ASA (4 g/day) OD + enema 1 g/day 5-ASA (4 g/day) BD + enema 1 g/day	Clinical and endoscopic remission at w 8 (UCDAI score <1)	52.1% 41.8% (*p* = 0.14)	Combined with 5-ASA enema, prolonged-release 5-ASA OD 4 g is as effective as 2 g twice daily for inducing remission
D’Haens et al. (2017) [44]	Randomized non-inferiority trial	817	5-ASA (3.2 g) OD 5-ASA (3.2 g) BD	Clinical and endoscopic remission at w 8 (MCS ≤ 2 with no individual score >1)	22.4% 24.6% (*p* = 0.005)	3.2 mg 5-ASA OD is non-inferior to a BD regimen
Sandborn et al. (2010) [45]	Randomized non-inferiority trial	1023	5-ASA (1.6–2.4 g/day) OD 5-ASA (1.6–2.4 g/day) BD	Clinical remission (SCCAI score ≤2 points) at mo. 6	90.5% 91.8% (*p* = 0.05)	OD dosing of delayed-release 5-ASA is as effective as BD dosing for maintenance of clinical remission
Kamm et al. (2007) [46]	RCT	343	MMX 5-ASA 2.4 g/day OD MMX 5-ASA 4.8 g/day OD Delayed-release oral 5-ASA 2.4 g/day (3 divided doses) Placebo	Proportion of patients in clinical and endoscopic remission (modified UCDAI <1 with rectal bleeding and stool frequency scores of 0, no mucosal friability, and a >1-point reduction in sigmoidoscopy score from baseline) at w 8	40.5% (*p* = 0.01) 41.2% (*p* = 0.007) 32.6% (*p* = 0.124) 22.1%	OD MMX 5-ASA 2.4 or 4.8 g/day are both superior to placebo in the induction of clinical and endoscopic remission
Lichtenstein et al. (2007) [47]	RCT	280	MMX 5-ASA 2.4 g/day BD MMX 5-ASA 4.8 g/day OD Placebo	Clinical and endoscopic remission (modified UCDAI score <1, with a score of 0 for rectal bleeding and stool frequency, and at least a 1-point reduction in sigmoidoscopy score) at w 8	34.1% (*p* < 0.01) 29.2% 12.9%	BD and OD MMX 5-ASA are efficacious for the induction of clinical and endoscopic remission
Kane et al. (2012) [48]	Phase IV multicentre open label	290	MMX 5-ASA 2.4 g/day OD	Clinical recurrence (defined as ≥4 bowel movements per day above the patient’s normal frequency and which were associated with any of the following symptoms: urgency, abdominal pain, or rectal bleeding) at mo. 6	23.5%	MMX 5-ASA 2.4 g/day OD is effective for maintaining quiescence
D’Albasio et al. (1997) [49]	RCT	69	5-ASA tablets (1.6 g/day) and 5-ASA enemas (4 g/100 mL) twice weekly 5-ASA (1.6 g/day) and placebo enemas/twice weekly	Maintenance of remission (mild symptoms and normal endoscopic appearance of mucosa) at mo. 12	39% 69% (*p* = 0.036)	5-ASA given daily by oral route and intermittently by topical route can be more effective than oral therapy alone.
Yokoyama et al. (2007) [50]	RCT	24	Weekend 5-ASA enema group (1 g 5-ASA enemas in the weekend plus oral 5-ASA 3 g/day for 7 days) Daily oral 5-ASA use only group (only oral 5-ASA 3 g/day for 7 days)	Incidence of relapse (as a score of ≥6 in clinical activity index and ≥3 in the endoscopic index)	18.2% 76.9% (multivariate HR: 0.19, 95% CI, 0.04–0.94)	Adding weekend 1 g 5-ASA enema to daily 3 g oral 5-ASA as maintenance therapy
Hanauer et al. (2007) [51]	RCT	301	5-ASA 2.4 g/day 5-ASA 4.8 g/day	Overall improvement (defined as complete remission or response to therapy) from baseline to w 6	57% 72% (*p* = 0.0384)	4.8 g/day dose may enhance treatment success rates in patients with moderate disease compared with mesalamine 2.4 g/day
Hanauer et al. (2005) [52]	RCT	386	5-ASA 2.4 g/day 5-ASA 4.8 g/day	Overall improvement (defined as either complete remission or a clinical response to therapy) from baseline to w 6	59% 72% (*p* = 0.036)	4.8 g/day dose results in significantly higher rates of overall improvement in patients with moderate disease compared with 2.4 g/day
Hiwatashi et al. (2011) [53]	RCT	123	5-ASA 4 g/day (2 divided doses) 5-ASA 2.25 g/day (3 divided doses)	UCDAI score before and after 8 weeks of treatment	3.0 (95% CI −3.8 to −2.3) 0.8 (95% CI −1.8 to 0.1)	4 g/day results in a significantly superior change in UCDAI score compared with 2.25 g/day
Pica et al. (2015) [54]	RCT	112	5-ASA 4.8 g 5-ASA 2.4 g	Maintenance of remission (defined as the absence of symptoms and the endoscopically documented absence of the inflammatory changes typical of active UC) at mo. 12	75% 64.2% (*p* = 0.3)	A daily dose of 4.8 g oral mesalamine results in increased rates and duration of remission compared to 2.4 g, in patients younger than 40 years and/or with extensive disease

Abbreviations: BD, bis in die (twice daily); MCS, Mayo Clinic Score; HR, hazard ratio; MX, multimatrix system; mo., month; OD, once daily; RCT, randomized controlled trial; SCCAI, Simple Clinical Colitis Activity Index; UC, ulcerative colitis; UCDAI, UC disease activity index; w, week; yr., year; CI, confidence interval.

**Table 2 jcm-09-02905-t002:** Studies reporting the efficacy of budesonide multimatrix system (MMX) in patients with mild-to-moderate ulcerative colitis.

Author (Year)	Study Design	Number of Patients	Study Arms	Primary Outcome	Results	Conclusions
Sandborn et al. (2012) [63]	RCT	509	Budesonide MMX 9 mg Budesonide 6 mg Mesalamine 2.4 g Placebo	Combined clinical and endoscopic remission (UCDAI score ≤1 point, with sub-scores of 0 for both rectal bleeding and stool frequency, no mucosal friability on colonoscopy, and a ≥1-point reduction from baseline in the endoscopic index score) at w 8	17.9% (*p* = 0.0143) 13.2% (*p* = 0.1393) 12.1% (*p* = 0.2200) 7.4%	Budesonide MMX 9 mg is safe and more effective than placebo in inducing remission
Travis et al. (2014) [64]	RCT	410	Budesonide MMX 9 mg Budesonide MMX 6 mg Budesonide 9 mg Placebo	Combined clinical and endoscopic remission (UCDAI score ≤1, with a rectal bleeding score of 0, stool frequency score of 0, mucosal appearance score of 0 and a ≥1-point reduction in baseline endoscopic index score) at w 8	17.4% (*p* = 0.0047) 8.3% (*p* > 0.05) 12.6% (*p* = 0.0481) 4.5%	Budesonide MMX 9 mg is safe and more effective than placebo in inducing combined clinical and endoscopic remission
Rubin et al. (2017) [67]	RCT	510	Budesonide MMX 9 mg Placebo	Combined clinical and endoscopic remission (UCDAI score of ≤1, with subscale scores of 0 for rectal bleeding, stool frequency, and mucosal appearance) at w 8	13.0% (*p* = 0.049) 7.5%	Budesonide MMX is safe and efficacious for inducing clinical and endoscopic remission for mild-to-moderate UC refractory to oral mesalamine therapy
Maconi et al. (2019) [69]	Retrospective cohort study	82	Budesonide MMX	Clinical remission (pMayo of 0–1 with a rectal bleeding sub-score = 0) at mo. 2	50%	Budesonide MMX is safe and effective in patients with mild disease activity
Danese et al. (2019) [70]	Prospective cohort study	326	Cohort 1: budesonide MMX + 5-ASA at least 14 days after increased/optimized 5-ASA dose Cohort 2: budesonide MMX + 5-ASA within 14 days since 5-ASA increased/optimized or without 5-ASA dose modification Cohort 3: budesonide MMX as monotherapy	Clinical benefit (≥3 point reduction UCDAI clinical sub-score) at the end of induction treatment	64.3% (*p* = 0.0096) 62.1% 33.3%	Budesonide is safe and well tolerated in about 60% of mild-to-moderate UC patients, in a real-life setting

Abbreviations: MMX, multimatrix system; mo., month; pMayo, partial Mayo Clinic Score; RCT, randomized controlled trial, UC, ulcerative colitis; UCDAI, UC disease activity index; w, week.

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
