# Peer review of "Key Strategies to Optimize Outcomes in Mild-to-Moderate Ulcerative Colitis"

_jcm, 2020, doi:10.3390/jcm9092905_

Round 1
Reviewer 1 Report
Solitano et al. reviewed the issue on Key strategies to optimize outcomes in mild-to-2 moderate ulcerative colitis. This review conducted the important for optimize UC patients outcome in mild to moderate disease activity. Authors have pointed out several important issues discussed so far by other group studies. Also, this review has been well written and logically addressed along with appropriate literatures.
Minor issues need to be revised as below.
1. Numbering is wrong at the line 121-122 in page 3. It should be changed 3.1. to 4.1. as well as 3.1.1. to 4.1.1.
2. Please indicate supplementary materials (Figure S1, Table S1, and Video S1) in the text. Although supplementary figures or tables which were not showing in the main figures, authors should indicate these information in the text.
Author Response
Comments from reviewer 1:
Solitano et al. reviewed the issue on Key strategies to optimize outcomes in mild-to-moderate ulcerative colitis. This review conducted the important for optimize UC patients outcome in mild to moderate disease activity. Authors have pointed out several important issues discussed so far by other group studies. Also, this review has been well written and logically addressed along with appropriate literatures.
Reply: We gratefully thank the reviewer for the positive comment.
Minor issues need to be revised as below.
- Numbering is wrong at the line 121-122 in page 3. It should be changed 3.1. to 4.1. as well as 3.1.1. to 4.1.1.
Reply: We have made the appropriate changes.
- Please indicate supplementary materials (Figure S1, Table S1, and Video S1) in the text. Although supplementary figures or tables which were not showing in the main figures, authors should indicate these information in the text.
Reply: We thank the reviewer for this comment. However, supplementary materials are not included in this paper.

Reviewer 2 Report
General statement:
The submitted paper on Key strategies to optimize outcomes in mild-to-moderate ulcerative colitis by Virginia Solitano et al. is a nicely written review mainly on the optimization of the two main anti-inflammatory substances mesalamine and budesonide when used for the treatment of patients with mild-to-moderate active ulcerative colitis (UC). It provides a sound and profound overview, referring to recent and previous mile stone studies in which the optimized use of these drugs and the use of generically engineered drugs (MMX technique, etc..) has led to better outcomes in patients with mild-to-moderate ulcerative colitis (UC).
In addition, the authors address the fact that by far less than half of all patients with mild-to moderate UC achieve long-term clinical and colonoscopic remission, when using mesalamine and budesonide. To overcome these drawbacks and challenges they propose
- the use of optimized mesalamine and budesonide formulations (it is noteworthy that one author is an employee for Ferring, a company selling these mesalazine and budesonide products, Pentasa and Cortiment)
- the use of modern immunomodulators (biologics and tofacitinib) earlier in the course of disease, avoiding long-term use of systemic steroids while they warn of the organo- and genotoxicity of purine analogues.
- the use of more intelligent diagnostic approaches (pan-colonoscopy, fecal calprotectin) and
- the use of Shared decision making and patient involvement, which obviously leads to better results.
They conclude with an elaborated algorithm in which both, conservative as well as modern physicians can choose their preferred mode of action, depending on how they define a relapse and the risk of colectomy.
The manuscript comprises an interesting discussion on status quo of oral and topical mesalamine and steroids and reminds of the fact that beyond these therapies, patients who are still considered to have mild to moderate disease do need other strategies unless they want to be stuck in the vicious circle of on-and-off steroids that eventually ends in colectomy in at least one out of ten patients.
Finally, using the format of chapters and subtitles makes the paper easy to read and comprehend.
Major criticism:
The submitted manuscript by Virginia Solitano, et al. promises to report on all strategies that could optimize the outcomes of patients with mild-to-moderate ulcerative colitis. The manuscript however, mainly, deeply and extensively concentrates on the optimization of the use of two compounds, mesalazine and budesonide, though in an excellent survey. When reading such an article, I and probably other physicians would also expect information on what to do, when the even most sophisticated use of these two drug systems fails to bring or keep these patients in remission.
Without this information, the here presented strategy implies that by an improved and optimized application of mesalamine and budesonide, more intelligent diagnostic techniques and better involvement of the patient, most patients with mild-to-moderate UC can be treated sufficiently and steroid side effects avoided, which is often not the case. Most experts and even less experienced physicians treating IBD patients are all well aware, that without immunomodulation even by optimizing all basic treatment strategies, steroid side effects are still a problem and by far less than half of all patients with mild-to moderate UC achieve long-term remission, as defined by modern standards.
The authors have obviously recognized this problem and tried to address it by adding the short chapter 4.3. on Patient stratification for early intervention in which they propose the consideration of early exposure to biologics or small molecules in patients with predictors of a more aggressive disease course. Looking at those predictors like young age at diagnosis, extensive disease, extra-intestinal manifestations, presence of endoscopic signs of disease activity (e.g. ulcers), and elevated inflammatory biomarkers (FC and C-reactive protein), most IBD physicians are well aware of the fact, that one or several of those symptoms are present in most of their patients with moderate UC. The authors’ suggestion to intervene early does furthermore exclude those patients in whom a late intervention would be appropriate as well; to date, those “late-intervention” patients comprise the majority of UC-patients in whom immunomodulation is started.
I my opinion, a review on key strategies to optimize outcomes in mild-to-moderate ulcerative colitis has to address the transition phase from mesalazine/budesonide to immunomodulation (why, when, who, how, what, how long?) in a separate and additional chapter. And what do the authors mean by careful evaluation of both, negative prognostic factors at diagnosis and the potential risks of overtreatment? Should we leave it at the discretion of the physician to treat a young patient with systemic steroid for many weeks or should we prompt him or her to start immunomodulation? The impact of undertreatment should be discussed as well. With such a background of the senior authors, one has also the chance to take a position: Fight inflammation, get rid of steroids, and avoid structural damage, for the sake of the patients’ gut and well-being; plus, there are published data to underscore these positions.
Indeed, I also miss a clear statement, e.g. extra chapter, on the (not-) use of systemic steroids. After so many years with biologics available, after so many trials with steroid-free remission being an accepted and important endpoint, after all the evidence on deleterious and disastrous side effects of steroids, the use of systemic steroids other than for a very short period of time has to be banned in IBD. Nevertheless, huge amounts of systemic steroids are still used worldwide in the majority of patients with mild-to-moderate UC. Hence, how should the key strategy be to get over this major issue?
I suggest that the authors change their manuscript accordingly and provide also key strategies to optimize outcomes beyond mesalazine and steroids by addressing all the above mentioned points.
Minor: Chapter 4: Correct numbering please.
Author Response
Comments from reviewer 2:
General statement:
The submitted paper on Key strategies to optimize outcomes in mild-to-moderate ulcerative colitis by Virginia Solitano et al. is a nicely written review mainly on the optimization of the two main anti-inflammatory substances mesalamine and budesonide when used for the treatment of patients with mild-to-moderate active ulcerative colitis (UC). It provides a sound and profound overview, referring to recent and previous mile stone studies in which the optimized use of these drugs and the use of generically engineered drugs (MMX technique, etc..) has led to better outcomes in patients with mild-to-moderate ulcerative colitis (UC).
In addition, the authors address the fact that by far less than half of all patients with mild-to moderate UC achieve long-term clinical and colonoscopic remission, when using mesalamine and budesonide. To overcome these drawbacks and challenges they propose:
- the use of optimized mesalamine and budesonide formulations (it is noteworthy that one author is an employee for Ferring, a company selling these mesalazine and budesonide products, Pentasa and Cortiment)
- the use of modern immunomodulators (biologics and tofacitinib) earlier in the course of disease, avoiding long-term use of systemic steroids while they warn of the organo- and genotoxicity of purine analogues.
- the use of more intelligent diagnostic approaches (pan-colonoscopy, fecal calprotectin) and
- the use of Shared decision making and patient involvement, which obviously leads to better results.
They conclude with an elaborated algorithm in which both, conservative as well as modern physicians can choose their preferred mode of action, depending on how they define a relapse and the risk of colectomy.
The manuscript comprises an interesting discussion on status quo of oral and topical mesalamine and steroids and reminds of the fact that beyond these therapies, patients who are still considered to have mild to moderate disease do need other strategies unless they want to be stuck in the vicious circle of on-and-off steroids that eventually ends in colectomy in at least one out of ten patients.
Finally, using the format of chapters and subtitles makes the paper easy to read and comprehend.
Reply: We gratefully thank the reviewer for the positive comment.
Major criticism:
The submitted manuscript by Virginia Solitano, et al. promises to report on all strategies that could optimize the outcomes of patients with mild-to-moderate ulcerative colitis. The manuscript however, mainly, deeply and extensively concentrates on the optimization of the use of two compounds, mesalazine and budesonide, though in an excellent survey. When reading such an article, I and probably other physicians would also expect information on what to do, when the even most sophisticated use of these two drug systems fails to bring or keep these patients in remission.
Without this information, the here presented strategy implies that by an improved and optimized application of mesalamine and budesonide, more intelligent diagnostic techniques and better involvement of the patient, most patients with mild-to-moderate UC can be treated sufficiently and steroid side effects avoided, which is often not the case. Most experts and even less experienced physicians treating IBD patients are all well aware, that without immunomodulation even by optimizing all basic treatment strategies, steroid side effects are still a problem and by far less than half of all patients with mild-to moderate UC achieve long-term remission, as defined by modern standards.
The authors have obviously recognized this problem and tried to address it by adding the short chapter 4.3. on Patient stratification for early intervention in which they propose the consideration of early exposure to biologics or small molecules in patients with predictors of a more aggressive disease course. Looking at those predictors like young age at diagnosis, extensive disease, extra-intestinal manifestations, presence of endoscopic signs of disease activity (e.g. ulcers), and elevated inflammatory biomarkers (FC and C-reactive protein), most IBD physicians are well aware of the fact, that one or several of those symptoms are present in most of their patients with moderate UC. The authors’ suggestion to intervene early does furthermore exclude those patients in whom a late intervention would be appropriate as well; to date, those “late-intervention” patients comprise the majority of UC-patients in whom immunomodulation is started.
Reply: We thank the reviewer for the remarkable comment. We believe that physicians should be prompt to evaluate when a patient with mild-to-moderate UC requires more intensive initial therapy in order to avoid getting stuck in the vicious circle of “on-and-off” steroids and to prevent structural damage and steroid-related adverse events. This consideration does not imply that a “late intervention” would not be appropriate as well in the majority of patients. Even though a definition of early intervention in UC is still lacking, selecting those patients who are at risk of disease progression might be a valid strategy in daily clinical practice. We have clarified our point in chapter 4.3 “Patient stratification for early intervention”.
In my opinion, a review on key strategies to optimize outcomes in mild-to-moderate ulcerative colitis has to address the transition phase from mesalazine/budesonide to immunomodulation (why, when, who, how, what, how long?) in a separate and additional chapter. And what do the authors mean by careful evaluation of both, negative prognostic factors at diagnosis and the potential risks of overtreatment? Should we leave it at the discretion of the physician to treat a young patient with systemic steroid for many weeks or should we prompt him or her to start immunomodulation? The impact of undertreatment should be discussed as well. With such a background of the senior authors, one has also the chance to take a position: Fight inflammation, get rid of steroids, and avoid structural damage, for the sake of the patients’ gut and well-being; plus, there are published data to underscore these positions.
Reply: We thank the reviewer for the brilliant analysis. As we discussed in our paper “Early Intervention in Ulcerative Colitis: ready for prime time?” (doi.org/10.3390/jcm9082646) published in the same Special Issue of JCM, UC is an evolving disease that can potentially lead to the development of progressive damage. Even though large randomized prospective studies are needed to assess the efficacy of early intervention strategy in ulcerative colitis and to identify predictors of complicated disease, we encourage physicians to raise the bar in terms of therapeutic goals, aiming at “disease clearance” as an ultimate achievable objective (symptomatic remission based on patient reported outcomes and mucosal healing, including both endoscopic and histological remission). We have explicitly stated our position in chapter 4.3 “Patient stratification for early intervention”.
Indeed, I also miss a clear statement, e.g. extra chapter, on the (not-) use of systemic steroids. After so many years with biologics available, after so many trials with steroid-free remission being an accepted and important endpoint, after all the evidence on deleterious and disastrous side effects of steroids, the use of systemic steroids other than for a very short period of time has to be banned in IBD. Nevertheless, huge amounts of systemic steroids are still used worldwide in the majority of patients with mild-to-moderate UC. Hence, how should the key strategy be to get over this major issue?
I suggest that the authors change their manuscript accordingly and provide also key strategies to optimize outcomes beyond mesalazine and steroids by addressing all the above mentioned points.
Reply: We perfectly agree with the reviewer. Therefore, we specified in chapter 5 the recommended use of systemic steroids in order to avoid the deleterious and prolonged use of steroids in the context of moderate UC. A correct regimen is prednisolone 40 mg/day (or Corticosteroid Dose Equivalents) for 1 week, lowering the daily dose by 5 mg each week, without exceeding the duration of an 8-week course. Moreover, during maintenance phase the use of thiopurines as steroid-sparing agents is sustained by both evidence and experience. Monotherapy and combo-therapy with anti-TNF agents represent feasible options. In addition, an increased awareness of the most updated evidence and a greater familiarity with evolving therapeutic goals and strategies by physicians seem to be valid key strategies to overcome this major issue.
Minor: Chapter 4: Correct numbering please.
Reply: We have made the appropriate changes.

Round 2
Reviewer 2 Report
Comment to the authors reply:
I see that the authors have address most of my concerns, but I would still want them to at least discuss the points that I mentioned:
- The fact that even by an improved and optimized application of mesalamine and budesonide, and the use of intelligent diagnostic techniques and better involvement of the patient, many patients with mild-to-moderate UC cannot be treated sufficiently and less than half of all patients with mild-to moderate UC achieve long-term remission, as defined by modern standards. Most experts and physicians treating IBD patients are all well aware, that those patients, even with a prudent use of systemic steroids, cannot be treated without immunomodulation.
@ 4.3. Patient stratification for ......:
- I think that using "earlier" instead of "early" would be more appropriate.
- Looking at predictors for an earlier use like young age at diagnosis, extensive disease, extra-intestinal manifestations, presence of endoscopic signs of disease activity (e.g. ulcers), and elevated inflammatory biomarkers (FC and C-reactive protein), in most of our patients with moderate UC one or several of those symptoms are present.
I have learned that the authors had submitted a paper on the subject of earlier and consistent anti-inflammatory therapy in UC, which had obviously been accepted. To underline the historic and ongoing reluctance of physicians to use effective drugs earlier, authors should also cite earlier publications like a review from 2011 (Current misunderstandings in the management of ulcerative colitis of Gut. 2011 Sep;60(9):1294-9) in addition to ref. 74 in 4.3.
Finally, the statement about thiopurines on 338-342 is in some contrast to the justified concerns about the limited therapeutic impact and the toxicity of thiopurines on lines76-79. The authors could repeat such a sentence after line 342, e.g. like "However, as mentioned above, the use of thiopurines is affected by it's limited therapeutic effect and it's toxicity and associated cancer risk (Thiopurines in Inflammatory Bowel Disease: New Findings and Perspectives. DJ Crohns Colitis. 2018 Apr 27;12(5):610-620)
Author Response
I see that the authors have address most of my concerns, but I would still want them to at least discuss the points that I mentioned:
The fact that even by an improved and optimized application of mesalamine and budesonide, and the use of intelligent diagnostic techniques and better involvement of the patient, many patients with mild-to-moderate UC cannot be treated sufficiently and less than half of all patients with mild-to moderate UC achieve long-term remission, as defined by modern standards. Most experts and physicians treating IBD patients are all well aware, that those patients, even with a prudent use of systemic steroids, cannot be treated without immunomodulation.
Reply: We have appreciated the reviewer’s comment. We further discussed this point as follows: “However, about half of the patients do not achieve sustained remission after optimization with 5-ASA and budesonide. Thus immunomodulators are still necessary to avoid the excessive use of systemic steroids and to maintain corticosteroid-free remission”.
@ 4.3. Patient stratification for ......:
I think that using "earlier" instead of "early" would be more appropriate.
Reply: We made the recommended change.
Looking at predictors for an earlier use like young age at diagnosis, extensive disease, extra-intestinal manifestations, presence of endoscopic signs of disease activity (e.g. ulcers), and elevated inflammatory biomarkers (FC and C-reactive protein), in most of our patients with moderate UC one or several of those symptoms are present.
Reply: We thank the reviewer for the comment. We have clarified our opinion suggesting that the early use of biological therapy should be preferred in patients with several red flags to prevent occurrence of negative outcomes and disease progression. Although the risk-stratification should be based on the above-mentioned predictors, the clinical decisions should be individually made by the physicians according to an individualized approach.
I have learned that the authors had submitted a paper on the subject of earlier and consistent anti-inflammatory therapy in UC, which had obviously been accepted. To underline the historic and ongoing reluctance of physicians to use effective drugs earlier, authors should also cite earlier publications like a review from 2011 (Current misunderstandings in the management of ulcerative colitis of from D'Haens et al. Gut. 2011 Sep;60(9):1294-9) in addition to ref. 74 in 4.3.
Reply: We have added this reference as requested.
Finally, the statement about thiopurines on 338-342 is in some contrast to the justified concerns about the limited therapeutic impact and the toxicity of thiopurines on lines76-79. The authors could repeat such a sentence after line 342, e.g. like "However, as mentioned above, the use of thiopurines is affected by it's limited therapeutic effect and it's toxicity and associated cancer risk (Thiopurines in Inflammatory Bowel Disease: New Findings and Perspectives. De Boer NKH, Peyrin-Biroulet L, Jharap B, Sanderson JD, Meijer B, Atreya I, Barclay ML, Colombel JF, Lopez A, Beaugerie L, Marinaki AM, van Bodegraven AA, Neurath MF.de Boer NKH, et al. J Crohns Colitis. 2018 Apr 27;12(5):610-620)
Reply: We thank the reviewer for this valid suggestion. We have made the appropriate change as requested.
